# A Statistical and AI Analysis of the Frequency Spectrum in the Measurement of the Center of Pressure Track in the Seated Position in Healthy Subjects and Subjects with Low Back Pain

**DOI:** 10.3390/s24103011

**Published:** 2024-05-09

**Authors:** Jan Jens Koltermann, Philipp Floessel, Franziska Hammerschmidt, Alexander C. Disch

**Affiliations:** 1Sport Medicine and Rehabilitation, Faculty of Medicine Carl Gustav Carus, Technische Universität Dresden, Fetscherstrasse 74, 01307 Dresden, Germanyfranziska.hammerschmidt@tu-dresden.de (F.H.); 2University Center for Orthopedics, Trauma & Plastic Surgery—University Comprehensive Spine Center (UCSC), Faculty of Medicine Carl Gustav Carus, Technische Universität Dresden, Fetscherstrasse 74, 01307 Dresden, Germany; alexander.disch@uniklinikum-dresden.de

**Keywords:** balance board, COP, power density, AI

## Abstract

Measuring postural control in an upright standing position is the standard method. However, this diagnostic method has floor or ceiling effects and its implementation is only possible to a limited extent. Assessing postural control directly on the trunk in a sitting position and consideration of the results in the spectrum in conjunction with an AI-supported evaluation could represent an alternative diagnostic method quantifying neuromuscular control. In a prospective cross-sectional study, 188 subjects aged between 18 and 60 years were recruited and divided into two groups: “LowBackPain” vs. “Healthy”. Subsequently, measurements of postural control in a seated position were carried out for 60 s using a modified balance board. A spectrum per trail was calculated using the measured CoP tracks in the range from 0.01 to 10 Hz. Various algorithms for data classification and prediction of these classes were tested for the parameter combination with the highest proven static influence on the parameter pain. The best results were found in a frequency spectrum of 0.001 Hz and greater than 1 Hz. After transforming the track from the time domain to the image domain for representation as power density, the influence of pain was highly significant (effect size 0.9). The link between pain and gender (*p* = 0.015) and pain and height (*p* = 0.012) also demonstrated significant results. The assessment of postural control in a seated position allows differentiation between “LowBackPain” and “Healthy” subjects. Using the AI algorithm of neural networks, the data set can be correctly differentiated into “LowBackPain” and “Healthy” with a probability of 81%.

## 1. Introduction

In everyday life, the muscles of the lower extremities and trunk, or rather their neuromuscular control, are relevant for balancing an upright posture. 

Well-developed core stability is extremely important for maintaining postural control and preventing falls. 

It is also a performance-determining factor in elite sport [1] and is considered as a key component for the occurrence or prevention of chronic back pain in the normal population at a physiological level [1,2,3,4,5].

Due to the fact that low back pain (LBP) is the most common musculoskeletal disorder worldwide and the cost burden on the healthcare system is expected to continue to increase in the coming decades, early detection of patients at risk of back pain is of particular importance [6,7].

Causes of chronic back pain are an altered neuromuscular activation and/or a reduced capacity of the spine-stabilizing muscles [8,9,10]. 

A frequently used measurement method for determining neuromuscular control is recording postural control in an upright standing position on a force plate [3,11]. Various parameters are used to quantify postural control. One of these variables is the center of pressure (CoP). In order to balance the body’s center of gravity, the CoP is in constant motion. Conclusions can be drawn about balance control from the distance traveled by the CoP during a measurement [12]. Analysis of the CoP while standing is also frequently used to detect possible deficits in neuromuscular trunk capacity [3,13]. However, the measurement of trunk stability in standing is controversial, as the CoP is significantly influenced by the performance of the lower extremities [14,15,16].

One possible approach to explicitly quantify neuromuscular control of the trunk-embracing muscles is recording postural control in a seated position [1,17]. This diagnostic method has the advantage that a distortion of the CoP due to physiological performance deficits or injuries to the lower limb can be excluded.

In general, various parameters can be extracted from the CoP, such as the path [18,19], the speed [20] and the area enclosed by it. The sagittal and lateral movements of the center of gravity are examined less frequently [21]. 

Frequency analysis or power spectral density (PSD) is an effective means of analyzing and displaying the sagittal and lateral movements of the center of gravity [22,23].

Different physiological causes can be attributed to the various resulting frequency spectra in the event of changes in the occurring frequency ranges [23,24]. A correlation between vestibular disorders and spectral changes in the range of 0.2 Hz, as well as the occurrence of proprioceptively induced fluctuations in the frequency band from 0.5 to 1.0 Hz has been demonstrated [12,25]. Various experiments by Kohen-Raz et al., 1996, also revealed vestibular disturbances. For this purpose, they examined the spectrum between 0 and 3 Hz, which is divided into eight frequency ranges. The method enables us to localize vestibular disturbances in the low-frequency range [26].

Proprioceptive or central disorders are localized in higher-frequency ranges [24,26]. An increased occurrence of frequencies below 0.1 Hz indicates a change in visual–labyrinthine control. On the other hand, an above-average occurrence of frequencies in the band from 0.2 to 1 Hz indicates deficits in the somatosensory control circuits, cerebellar damage, or mobilization of the stretch reflex [24]. Nashner et al., 1978, were the first research group that demonstrated an increase in movements with frequencies in the range of 0.1 to 0.3 Hz due to the loss of visual and somatosensory stimuli [22].

It is assumed that individual main frequency ranges can be assigned to specific causes of interference [27,28]. However, the results of this approach have not been evaluated in detail yet [29]. 

In addition to statistical methods, procedures from the field of AI are also to be tested. Tagliaferri et al. have presented a comprehensive review on the use of AI methods. To date, there have been no studies on the measurement of neuromuscular control in low back pain (LBP) in a seated position or the AI-supported evaluation of the frequency spectra of the CoP tracks. For AI methods to contribute to the classification of low back pain and to guide treatment allocation and monitor success, large data sets with known and exploratory clinical characteristics should be investigated. In addition, the reliability, validity and predictive ability of AI methods in low back pain and their ability to guide treatment allocation for improved patient outcomes need to be demonstrated [30].

Koltermann et al., 2020, examined the PSD in people with and without back pain while standing and were able to show that the areas in the PSD for back pain patients (Korff 4 or higher) are smaller than in subjects with a Korff of zero. This reduced power metabolic rate can be interpreted as an indication of restricted mobility (reduced neuromuscular control). This can provide information about the biomechanical cause of the complaints [25]. 

In addition, Barbado et al., 2016, were able to show that the CoP track, measured in a seated position, differs in symptom-free athletes from different sports. The authors concluded that different neuromuscular trunk capacities do exist in the differing cohorts [1]. 

However, it is unclear whether the CoP track also differs between patients with LBP with reduced trunk capacity and healthy adults in the sitting position.

Furthermore, in this context it should be examined whether the analysis of partial sections of the total PSD, which can be derived from the CoP track, with the aid of an AI-supported assessment, is a better predictor for the differentiation of back pain than the isolated consideration of the CoP track.

## 2. Materials and Methods

For this prospective study with a cross-sectional design, 188 subjects between the age of 18 and 60 were recruited. All subjects were examined in a sports medical department certified by the German Olympic Sports Confederation. The group of LBP subjects was significantly older than the healthy subjects. For this reason, age was not included as a parameter in the statistical analysis, nor was age taken into account for the training of the AI in order to rule out the possibility of the AI learning a pseudo-correlation.

The volunteers were only included if they answered “no” to all questions on the “Initial questionnaire to assess the health risk for people doing sport” of the German Society for Sports Medicine and Prevention. Back pain was assessed using Korff’s Chronic Pain Grade Questionnaire (Klasen et al., 2004 [31]).

The anthropometric data of the test subjects are summarized in the following table (Table 1).

As part of the quality assurance strategy, 45 test subjects were excluded in stages. In the first stage, all test subjects who were unable to maintain sitting position for more than 60 s or who performed excessive movements were excluded. In the second stage, all subjects with technically impossible results were excluded.

All values greater than 10^6^ were declared as outliers (implausible values). In addition, all values with a distance greater than 90% of the interquartile range from the interval of the middle 50% of the sample elements were excluded. 

Postural control was measured over 60 s using a balance board, which is based on a further technical development of a WII balance board (BB) [32]. The measurement setup and data processing are shown schematically in Figure 1.

The sampling rate was 1 kHz [33]. An NI USB 6001 converter from National Instruments with a sampling rate of 14 bits was used to convert the analog measurement data into a digital format [34]. On the software side, the raw data were recorded using LabVIEW 2014 from National Instruments (Austin, TX, USA).

The BB measurement data were processed using LabVIEW 2020. Before calculating the CoP curve, the raw data were filtered using a third-order Butterworth low-pass filter. The cut-off frequency was determined for the entire cohort according to the method of Koltermann et al., 2019 [35]. This ensures that the relationship between the anthropometry and the measurement variable is not changed [36]. After this procedure, the CoP curve was determined.

During the measurements, the test subjects had to sit in an active upright position in the middle of the balance board. They were also asked to look at a marker at a 3 m distance. During the tests, they were asked to keep their hands crossed in front of their chest. The legs should be kept parallel. The height of the test position was adjusted so that the test subjects’ legs were 50 cm above the floor. 

During the measurement, the test subjects were asked to keep their body position as still as possible and their trunk tension as high as possible (Figure 2). The tests were carried out by physiotherapists or research assistants. All those who carried out the measurements were instructed in the use of the technology and the procedure in advance of the study.

First, the measured CoP track in sitting was compared with 8 area sections over different frequency ranges of the LBP and healthy groups using a *T*-test. The results were used as a selection criterion for the subsequent parameter study using ANOVA.

A spectrum per trail was calculated from the measured CoP tracks using PSD analysis. In the range from 0.01 to 10 Hz, the spectrum was integrated in different areas to form the area below the envelope. These areas were included in the statistical analysis. To investigate whether individual parameters have an influence on the result of the CoP track or the area in the spectrum, an ANOVA was created using R 4.2.2 in RStudio and the power and effect strength were calculated.

Different algorithms for data classification and prediction of these classes were tested for the parameter combination with the highest statically proven influence on the parameter pain. For this purpose, support vector machines, trees, random forests, and neuronal models were compared. The training and evaluation of the models was performed with R 4.2.2 in RStudio. The libraries randomForest, rpat, e1071, and neuralnet were used to create the models. The data set were randomly split 70/30 into training data and evaluation data. All input parameters were normalized. The training and evolution process was repeated 10 times for each model and the results were summarized to a mean and standard deviation. A training and evaluation data set was generated for each run, which was the same for all models.

## 3. Results

The data collected from the CoP measurement in the seated position were converted into a spectrum using PSD analysis and then integrated in various ranges from 0.001 Hz to 10 Hz. The calculated areas under the spectrum were used in the subsequent analysis. Table 2 summarizes the calculated results of the test subjects and evaluates them using a *T*-test that differentiates the mean values between the LBP group and the healthy group.

Table 2 shows that the best results were found for the ranges of 0.001 Hz and greater than 1 Hz.

Table 3 shows the statistical evaluation of the different age groups depending on the affiliation of the pain status via the evaluated parameters such as CoP track and the frequency bands as mean value with standard deviation. The basic trend shows that the CoP track becomes longer with increasing age, but the performance turnover sings. To investigate why the ratio of the results from the pain group to the non-pain group is reversed from CoP track to the performance peak of PSD, five samples each were selected from the pain group and the healthy group and the acceleration was calculated for the CoP tracks at each point. The mean value of the maximum acceleration per movement cycle was then evaluated and the mean value calculated for the respective group. This resulted in an acceleration of 1.9 cm/s^2^ for the healthy group and 2.4 cm/s^2^ for the pain group. If you look at the plotted CoP tracks in the diagram, the healthy people show a more circular movement pattern. The pain patients show more of an elliptical track pattern (see schematic diagram, Figure 3).

Table 4 shows the correlation between size and gender in relation to the CoP track and the spectrum. Here, only the spectrum 0.001–10 Hz is shown as an example because it is representative of the other spectra. The plot shows the mean value and the standard deviation. The ANOVA in Table 5 and Table 6 shows that there is no statistically significant correlation between size, gender, and CoP track or spectrum.

Table 5 shows that of the parameters investigated, height, weight, and the combination of shoe size and weight.

These parameters all show a statistically significant influence, but the effect size of greater than 0.6 must be regarded as moderate.

Table 6 shows an ANOVA for the frequency spectrum (frequency range 0.001–10 [Hz]) in sitting and the influence of anthropometric data and the degree of pain. When looking at the individual variables, the link between the variables and the CoP spectrum is different from that of the original CoP track when the track is transformed from the time to the image range for display as power density. In this case, the influence of pain on the individual variables is highly significant with an equally high effect size of 0.9. The correlations between pain and gender (*p* = 0.015) and pain and height (*p* = 0.012) are also significant.

Table 7 shows that all AI methods can assign the test subjects to the correct class “LowBackPain” or “Healthy” with a probability of greater than 60%. The neural network performed best with a probability of 81.39%.

## 4. Discussion

Against the background of maintaining an active lifestyle and avoiding immobility, the quantification of postural control and the evaluation of the effect of certain exercises on the development of balance ability are of particular importance, especially for older people, stroke patients, patients with multiple sclerosis, and patients with chronic back pain [37,38]. Therefore, it is necessary to have a measurement method where the neuromuscular control of people with physical limitations can also be examined in a reliable, safe, and low-risk setting.

In addition, the quantification and differentiation of neuromuscular control of the trunk is relevant for optimizing athletic performance and injury prevention. Initial studies have shown that there are indeed sport-specific profiles of postural control in the standing and seated positions [1,17,39]. This leads to the conclusion that a specific assessment of the trunk-stabilizing muscles and postural control in specific cohorts appears to be useful in order to derive differentiated training programs and test their effect.

The aim of this study was to investigate whether parts of the spectrum, extracted from the total CoP track, are a better predictor for differentiating between “Healthy” and “LowBackPain” than the total CoP track by using an AI-supported analysis method.

For this purpose, the CoP track was determined and evaluated, as were different areas in the spectrum of the PSD of the CoP track. The statistical evaluation of different sections of the PSD showed that the sections 0.001–4 Hz and 0.001–10 Hz show significant differences in the power density of healthy people compared to LBP patients.

In the frequency range between 0.001 and 10 [Hz], LBP patients had higher power turnover than the healthy comparison group. However, the pain group had lower CoP tracks. The results could be due to reduced proprioception. Our data are consistent with the results of Lamoth et al., 2006, who investigated gait kinetics in LBP patients and found a more rigid and less variable kinematic coordination in the transverse plane and a less tight and more variable coordination in the frontal plane. They attributed the altered coordination to impaired neuromuscular control of the lumbar erector spinae and altered trunk coordination as a direct consequence of lumbar spine pain. This could account for both the higher power turnover and the lower CoP tracks in the acute study [40]. 

This assumption is supported by the results of Riadh Bizid et al., 2009, who investigated the effect of calf muscle fatigue with regard to a change in frequency during stance in top athletes. They found an increased power turnover in higher frequency ranges after the fatigue phase and attributed this to an altered contribution of proprioceptive information (myotatic loops) to the maintenance of postural control. This could also explain the increased power turnover in LBP patients in the current study. It is possible that the postural task in the present study leads to the LBP patients tiring more quickly and thus making higher proprioceptive efforts than the healthy subjects, which ultimately results in increased power turnover [12]. 

The influence of the length of the lower extremities on power expenditure in a seated position has not yet been researched. As this could serve as a counter pendulum, the influence of leg length should be investigated in future studies.

In order to be able to assess the differentiated influence of the various neuromuscular information systems (visual, vestibular, procedural) on postural control in a sitting position more precisely, future studies should be carried out on different groups of people with EMG recordings and with open and closed eyes. This would allow therapy programs for patients with different clinical pictures to be derived in a more targeted manner.

A further limitation is that the age structure of the healthy group is not the same as that of the pain group, as it is difficult to recruit older subjects without orthopedic findings. 

This includes various effective pendulum lengths such as the axis of movement around the hip, which is between 5 and 10 cm above the measuring plate in the seated position and thus causes movements in the frequency range between 1.5 and 3 Hz. Furthermore, movements in the area of the SIJ (approx. 20 cm corresponds to 1 Hz) and thoracic spine (approx. 30 to 45 cm corresponds to 0.9–0.7 Hz) can be differentiated. In addition, there are many other high-frequency movements that arise, for example, through the activation of the gluteal and thigh muscles. These have direct contact with the support surface and are intuitively included in the postural control loop. Taken individually, each area appears to be too small and individual to be able to derive a significant statement. Taken as a whole, however, it can be seen that covering different areas in the spectrum leads to a valid result. Against the background of the biomechanical interlinking of the individual moving axes, this appears comprehensible.

In the presented study, a gender-dependent significant difference was found for the CoP track. When the subjects were examined using the frequency spectrum, this difference was close to significance at 0.07. Women who were included in the study had a lower power metabolic rate than men, regardless of whether they had LBP or not. This suggests that the difference in muscle mass between men and women is proportionally reflected here [41]. At the same time, however, it was demonstrated, that women could better use their existing muscles. The results for the combination of the parameters pain and gender in relation to the power turnover in the spectrum show a significant *p*-value (0.015), but the power (0.69) is low. This is due to the very large dispersion of the “men with back pain” group. Another possible influence may be that the LBP subjects are older than the healthy subjects. However, an indication can be derived that the method as proposed by Koltermann et al., 2017, in the standing position is transferable to the measurement setup in the sitting position [32]. Based on these results, various algorithms from the field of artificial intelligence were applied to the data to classify data sets. Here, the neural networks stand out in particular with a prediction quality of 81%. The random forest and tree methods also show good predictive behavior.

## 5. Conclusions

Recording postural control in a seated position, women and men show different power conversion rates in the frequency range between 0.001 Hz and 10 Hz for mastering the set task. The same applies to the differentiation between the “LowBackPain” and “Healthy” groups. Here, the LBP group had higher power conversion rates than the pain-free cohort. This can be explained by a reduced ability to control the trunk-stabilizing muscles. Using neural networks, it is possible to correctly differentiate between the “LowBackPain” and “Healthy” groups from the data set, with a probability of 81%.

## Figures and Tables

**Figure 1 sensors-24-03011-f001:**
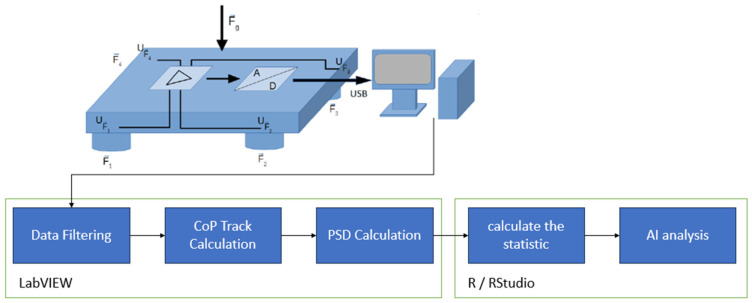
Schematic representation of measurement data acquisition and data flow through data processing.

**Figure 2 sensors-24-03011-f002:**
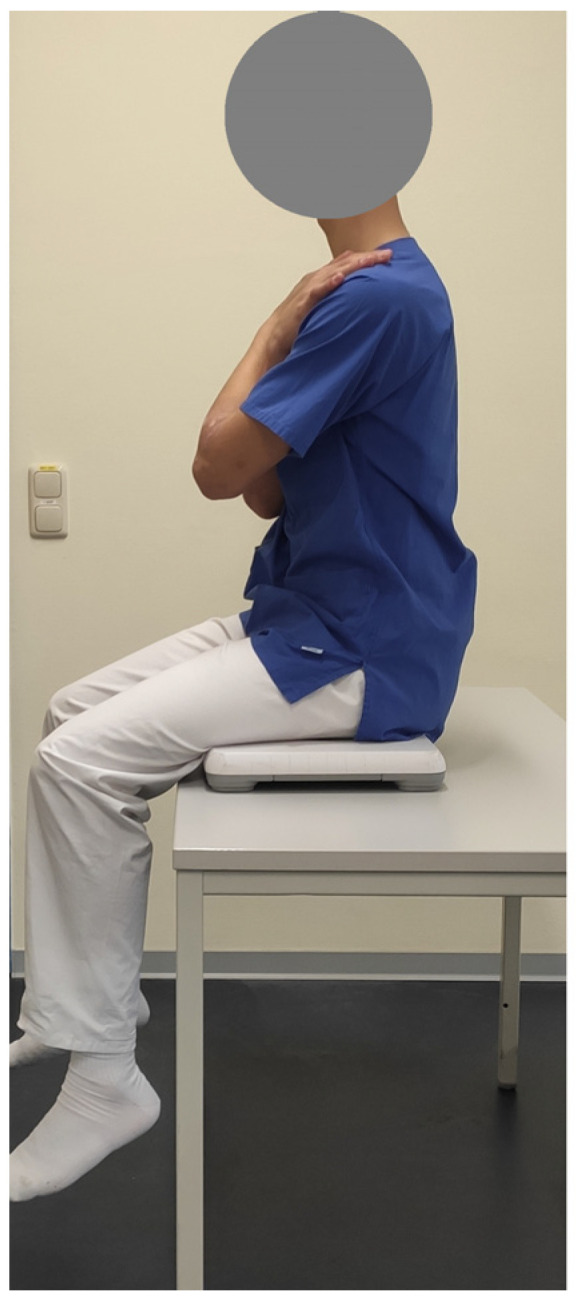
Test person in seated position on the balance board with an active upright upper body positioning during the measurements.

**Figure 3 sensors-24-03011-f003:**
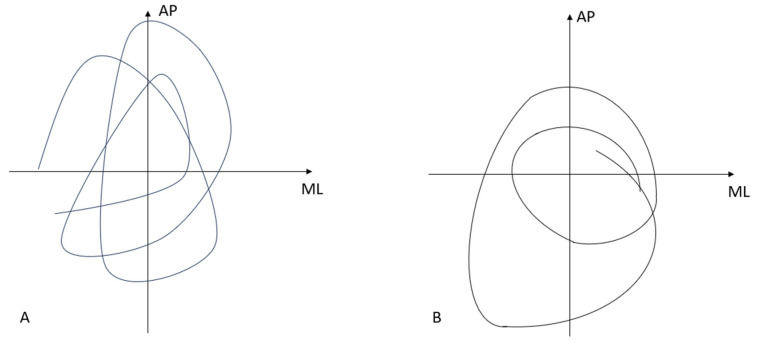
Schematic representation of the CoP track in (**A**) joke subjects and (**B**) healthy subjects.

**Table 1 sensors-24-03011-t001:** Anthropometry of the entirety of all subjects.

	Total Group	Male (52%)	Female (48%)
Age (Y.)	39.9 ± 17.5	40.4 ± 16.6	40.54 ± 17.66
Weight (kg)	75.8 ± 15.6	83.9 ± 13.7	66.13 ± 11.93
Size (cm)	171.5 ± 25.8	180.9 ± 7.7	160.30 ± 34.21
BMI (kg/m^2^)		25.6 ± 3.8	24.2 ± 4.5
Shoe size (EU)	42.8 ± 2.7	42.5 ± 5.5	38.9 ± 1.5

**Table 2 sensors-24-03011-t002:** Values for the various frequency intervals.

	No Pain	Pain	Difference	*p*-Value
CoP [cm]	91.636945	81.43684462	10.20010038	0.041478138
Frequency range in sitting position 0.001–10 [Hz]	0.025768105	0.114558354	−0.088790249	5.12899 × 10^−6^
Frequency range in sitting position 0.001–8 [Hz]	0.026194018	0.11441312	−0.088219102	5.75764 × 10^−6^
Frequency range in sitting position 0.001–6 [Hz]	0.025840127	0.114006704	−0.088166577	5.59621 × 10^−6^
Frequency range in sitting position 0.001–4 [Hz]	0.025204782	0.113160361	−0.087955579	5.31894 × 10^−6^
Frequency range in sitting position 0.001–1 [Hz]	0.074196032	0.104773996	−0.030577964	0.131960942
Frequency range in sitting position 1–8 [Hz]	0.008238026	0.008186706	5.13208 × 10^−5^	0.971152805
Frequency range in sitting position 1–4 [Hz]	0.007466393	0.007329342	0.000137051	0.915010431
Frequency range in sitting position 4–8 [Hz]	0.001292355	0.001484001	−0.000191646	0.549415161

**Table 3 sensors-24-03011-t003:** Statistical correlation between age, pain condition, and the CoP track or spectrum.

Age [a]	20–29		30–39		40–49		>50	
Pain	Yes	NO	YES	NO	YES	NO	YES	NO
CoP Track [cm]	85.29 (24.73)	89.64 (22.58)	70.34 (23.9)	92.19 (27.88)	77.45 (28.89)	114.37 (7.33)	86.84 (31.01)	100.41 (13.42)
0.001–10 Hz	0.11 (0.13)	0.02 (0.02)	0.12 (1.14)	0.03 (0.03)	0.12 (0.15)	0.02 (0.008)	0.09 (0.14)	0.02 (0.01)
0.001–8 Hz	0.108 (0.13)	0.024 (0.02)	0.106 (1.14)	0.033 (0.03)	0.123 (0.15)	0.025 (0.008)	0.095 (0.14)	0.023 (0.017)
0.001–6 Hz	0.108 (0.13)	0.024 (0.025)	0.106 (0.14)	0.032 (0.03)	0.123 (0.15)	0.025 (0.008)	0.094 (0.14)	0.023 (0.017)
0.001–4 Hz	0.108 (0.13)	0.023 (0.024)	0.105 (0.14)	0.031 (0.03)	0.122 (0.15)	0.024 (0.008)	0.093 (0.14)	0.022 (0.17)
1–4 Hz	0.0058 (0.005)	0.0067 (0.007)	0.0062 (0.006)	0.0097 (0.007)	0.0073 (0.007)	0.0105 (0.003)	0.0069 (0.011)	0.0073 (0.003)
1–8 Hz	0.0063 (0.005)	0.0074 (0.008)	0.0067 (0.007)	0.0105 (0.008)	0.0081 (0.007)	0.0113 (0.003)	0.0076 (0.001)	0.0082 (0.004)
4–8 Hz	0.0007 (0.0005)	0.0012 (0.002)	0.0007 (0.0004)	0.0013 (0.001)	0.0013 (0.001)	0.0013 (0.0001)	0.0014 (0.002)	0.0016 (0.001)
0.001–1 Hz	0.104 (0.12)	0.075 (0.11)	0.099 (0.13)	0.079 (0.08)	0.115 (0.14)	0.025 (0.003)	0.087 (0.13)	0.059 (0.07)
n	5	54	4	17	39	2	16	6

**Table 4 sensors-24-03011-t004:** Statistical correlation between size, gender, and the CoP track or spectrum.

Height [cm]	Gender	CoP Track [cm]	0.001–10 Hz	n
150	F	88.3 (30.4)	0.05 (0.06)	14
160	M	75.6 (24.5)	0.04 (0.06)	4
160	F	81.1 (24.6)	0.06 (0.11)	40
170	M	85.3 (24.6)	0.07 (0.07)	19
170	F	89.01 (26.23)	0.09 (0.19)	25
180	M	90.2 (30.7)	0.05 (0.05)	29
180	F	81.2 (8.45)	0.02 (0.01)	6
190	M	109.1 (20.9)	0.04 (0.01)	6

**Table 5 sensors-24-03011-t005:** ANOVA for pain vs. CoP track and anthropometry.

Term	Statistic	*p*-Value	Etasq	Omegasq	Power
Sex	5.928	0.016	0.037	0.03	0.683
Weight	5.474	0.021	0.034	0.027	0.648
Body size	0.957	0.33	0.006	0	0.165
Pain	0.659	0.419	0.004	−0.002	0.128
Sex: Weight	0.605	0.438	0.004	−0.002	0.122
Sex: Body size	5.077	0.026	0.031	0.025	0.616
Weight: Body size	0.13	0.72	0.001	−0.005	0.065
Sex: Pain	0.653	0.421	0.004	−0.002	0.128
Weight: Pain	1.813	0.181	0.011	0.005	0.27
Body size: Pain	1.202	0.275	0.007	0.001	0.195
Sex: Weight: Body size	3.153	0.078	0.019	0.013	0.427
Shoe size: Sex: Pain	3.591	0.061	0.022	0.016	0.474
Shoe size: Weight: Pain	0.08	0.778	0	−0.006	0.059
Sex: Weight: Pain	0.087	0.768	0.001	−0.006	0.06
Sex: Body size: Pain	0.419	0.519	0.003	−0.004	0.099
Weight: Body size: Pain	0.794	0.375	0.005	−0.001	0.145
Sex: Weight: Body size: Pain	0.047	0.829	0	−0.006	0.055

**Table 6 sensors-24-03011-t006:** ANOVA for pain vs. frequency spectrum (frequency range in sitting 0.001–10 [Hz]) in sitting and anthropometry.

Term	Statistic	*p*-Value	Etasq	Omegasq	Power
Sex	1.236	0.072	0.001	−0.004	0.064
Weight	3.632	0.054	0.002	−0.003	0.093
Body size	0.004	0.951	0	−0.005	0.05
Pain	42.458	0.0002	0.196	0.191	0.9
Shoe size: Weight	3.373	0.069	0.016	0.011	0.451
Sex: Weight	1.254	0.265	0.006	0.001	0.202
Sex: Body size	0.46	0.499	0.002	−0.002	0.104
Weight: Body size	1.615	0.206	0.007	0.003	0.246
Sex: Pain	6.063	0.015	0.028	0.023	0.692
Weight: Pain	2.91	0.091	0.013	0.009	0.4
Body size: Pain	6.466	0.012	0.03	0.025	0.72
Sex: Weight: Body size	5.054	0.027	0.023	0.019	0.614
Sex: Weight: Pain	3.333	0.071	0.015	0.011	0.447
Sex: Body size: Pain	0.746	0.39	0.003	−0.001	0.139
Weight: Body size: Pain	8.231	0.005	0.038	0.033	0.818
Sex: Weight: Body size: Pain	4.613	0.034	0.021	0.017	0.575

**Table 7 sensors-24-03011-t007:** Comparison of different Ki methods for data classification and their quality as an average value over 10 different training runs.

Type	Result [%]	Standard Deviation
SVM	60.46	±4.56
Tree	79.06	±5.64
Ranom Forest	74.47	±4.89
NN	81.39	±2.68

## Data Availability

The original contributions presented in the study are included in the article, further inquiries can be directed to the corresponding author/s.

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
