# Peer review of "A Statistical and AI Analysis of the Frequency Spectrum in the Measurement of the Center of Pressure Track in the Seated Position in Healthy Subjects and Subjects with Low Back Pain"

_sensors, 2024, doi:10.3390/s24103011_

Round 1
Reviewer 1 Report
Comments and Suggestions for Authors
The sample universe is greatly expanded, with very large variations between the indices surveyed.
It was also not possible to identify which other characteristics were kept fixed. For example, what type of pre-existing disease existed? The type of food? What kind of exercise or sedentary lifestyle was there? These cuts become important, otherwise the conclusion can only be referenced as a correlation, but not cause-effect.
I recommend that the results be better explored by subdividing the indices used for better data extraction and generating more appropriate conclusions. Also, in order for cause-effect to be more noticeable, it would be good to define in advance what kind of habits, etc., each subgroup has.
The results may even differentiate those who have it from those who do not have an adequate posture, but nothing can be explained about the causes. And that leaves the search weak from an application point of view. There is no procedural novelty, no innovation in the way data is captured, etc. It was a study of basic application of statistics, only.
Author Response
Thank you for your comments. we have made adjustments and additions on the basis of your comments and those of the other experts.
unfortunately, we did not collect psychosocial items on the test subjects. for this reason, we cannot make any statements on this. In order to obtain reliable results here, the sample would have to be much larger.
Reviewer 2 Report
Comments and Suggestions for Authors
The paper topic is of great interest, and the clinical trial of importance. In so far I can evaluate it, English writing is clear and appropriate.
I proposed anyway a major revision for the following reasons:
1. the clinical trial and the data processing is a big piece of work and it has to be published, nevertheless:
2. the authors have to go further in their discussion.
2.1. There is too many bibliography without direct link to the results. for example, the frequency domains are presented, but the link with the clinical results is missing. What could be deduced on the different frequency ranges from experimental data knowing that frequencies are related to neuromuscular groups?
2.2. Age is a big confusing factor in this study. When people are aging, the frequency domain changes. LPB patients have different signatures but they are olders. This must be commented in the discussion regarding appropriate litterature.
2.3. there is a coupling between gender and body size. Again, this should be a confounding factor, as women are smaller than men. Did this point checked? As a more general comment, there are few information on the LBP and shame group, and the statistically significant differences.
2.4. the set-up uses and unstable hemispheric hard cushion. What is the influence of the radius on the spectral signature of a patient/subject? Is it the same set-up for other results from the litterature?
3. Material and methods
3.1. Presentation of data: It is difficult to visualize the signal processing. I suggest to take more time to explain and to add a figure to present clearly the signal (or its frequency spectrum).
4. Results
4.1. There is an issue with the frequency range: is it reasonable to analyze frequencies from 0.001 Hz? the test duration is 60 s, so, taking into account Shannon Theorem, the lowest frequency should be 0.05 Hz
4.2. comments on table 3 and 4 should be revised. As an example, it is written "Table 3 shows that of the parameters examined, shoe size, weight ..." (line 185) but shoe size does not appear as a main influencing factor, and it is not statistically funded. (By the way, use dots instead of commas as a decimal separator).
5. General comment
The paper title emphases the use of AI, but is really the central topic of the paper that focuses more on frequency domains. In fact, what could we expect from AI classification here? Low back pain is not a hidden syndrome. Precisely, patient feel pain and there no interest in detecting LBP in the population. We need more an indicator to characterize it severity. Maybe the authors could comment on that or change their focus?
Author Response
thank you very much for the critical reading and questioning of our manuscript. we have gone through your comments and have also tried to implement them in the best possible way in the context of the other reviewers. You will find the changes below and in the manuscript.
The paper topic is of great interest, and the clinical trial of importance. In so far I can evaluate it, English writing is clear and appropriate.
I proposed anyway a major revision for the following reasons:
- the clinical trial and the data processing is a big piece of work and it has to be published, nevertheless:
we have expanded the data and statistics presented
- the authors have to go further in their discussion.
2.1. There is too many bibliography without direct link to the results. for example, the frequency domains are presented, but the link with the clinical results is missing. What could be deduced on the different frequency ranges from experimental data knowing that frequencies are related to neuromuscular groups?
This is very difficult to answer as there is little literature on the subject and our data has also shown that small sections cannot be linked so well.
2.2. Age is a big confusing factor in this study. When people are aging, the frequency domain changes. LPB patients have different signatures but they are olders. This must be commented in the discussion regarding appropriate litterature.
we have improved the text and analysis here
2.3. there is a coupling between gender and body size. Again, this should be a confounding factor, as women are smaller than men. Did this point checked? As a more general comment, there are few information on the LBP and shame group, and the statistically significant differences.
we have improved the text and analysis here
2.4. the set-up uses and unstable hemispheric hard cushion. What is the influence of the radius on the spectral signature of a patient/subject? Is it the same set-up for other results from the litterature?
we carried out various tests, including the moving background. however, only the data from the statically seated image was included in the evaluation. the image was inserted by mistake and has now been replaced with the correct one
- Material and methods
3.1. Presentation of data: It is difficult to visualize the signal processing. I suggest to take more time to explain and to add a figure to present clearly the signal (or its frequency spectrum).
we have added a schematic representation to the description.
Results
4.1. There is an issue with the frequency range: is it reasonable to analyze frequencies from 0.001 Hz? the test duration is 60 s, so, taking into account Shannon Theorem, the lowest frequency should be 0.05 Hz
in theory, you are absolutely right. in practice, however, it makes sense to choose a larger range in order to represent all the frequencies it actually contains. For this reason, we usually only exclude the equal anteul and specify the largest possible window. The algorithm is tolerant of such an approach.
4.2. comments on table 3 and 4 should be revised. As an example, it is written "Table 3 shows that of the parameters examined, shoe size, weight ..." (line 185) but shoe size does not appear as a main influencing factor, and it is not statistically funded. (By the way, use dots instead of commas as a decimal separator).
we have taken the hint and adjusted the tables and statements.
- General comment
The paper title emphases the use of AI, but is really the central topic of the paper that focuses more on frequency domains. In fact, what could we expect from AI classification here? Low back pain is not a hidden syndrome. Precisely, patient feel pain and there no interest in detecting LBP in the population. We need more an indicator to characterize it severity. Maybe the authors could comment on that or change their focus?
we have tried to adapt the title better to the paper. if you have any suggestions, we are open to them as well
Reviewer 3 Report
Comments and Suggestions for Authors
Dear Authors,
I enclose the pdf file of your submission which contains my minor comments.
Best,

Minor editing of English language required
Author Response
thank you very much for the critical reading and questioning of our manuscript. we have gone through your comments and have also tried to implement them in the best possible way in the context of the other reviewers. You will find the changes below and in the manuscript.
introduction
This section is rather long. Try to reduce the lenght by at least 20%.
we would like to refrain from shortening the introduction because, from our point of view, so many topics run into each other and these are important for different reader groups to be introduced to the topic
methodes
1) Did the authors use a clinical statement for observational studies like the STROBE statement? In more details, report on:
-Relevant dates of the research.
-Efforts used to address potential sources of bias.
2) Sample size calculation. yes 200
3) Who assessed the participants? The tests were carried out by physiotherapists or research assistants. All those who carried out the measurements were instructed in the use of the technology and the procedure in advance of the study.
results
Give also info on:
1) Harms description
2) Compliance to evaluations (acceptability).
could you elaborate a little more on this comment we are not sure how to interpret this.
discussion
A limitation section is recommended at the end of Discussion.
is added
Round 2
Reviewer 1 Report
Comments and Suggestions for Authors
I am satisfied with the revision of the text. I would only adapt Figure 1 to be better diagrammed on the lateral limits of the paper dimensions.
Author Response
we are pleased that the changes have led to a significant improvement. thank you for your efforts.
we have also adjusted the image text width
Reviewer 2 Report
Comments and Suggestions for Authors
the reviewer would like to thank the authors for their efforts to comply with the comments.
May I add few typos in the new text:
* line 196: CoP track (instead of Cop)
* table 4 (title): 0.001-10 Hz (instead of 0,001-10 Hz )
* line 216: "Here only the spectrum 0.01-10Hz " this is not consistent with table 4. It would be rather "Here only the spectrum 0.001-10Hz "
* Table 6: there are still commas instead of dots
* line 265: "LBP patients" insead of "lbp patients"
* line 273: "the lower COP tracks " instead of "the lower cop tracks "
Author Response
we are pleased that the changes have led to a significant improvement. thank you for your efforts.
we have incorporated the minor comments to the text